# Is Biochar from the Torrefaction of Sewage Sludge Hazardous Waste?

**DOI:** 10.3390/ma13163544

**Published:** 2020-08-11

**Authors:** Andrzej Białowiec, Jakub Pulka, Marzena Styczyńska, Jacek A. Koziel, Joanna Kalka, Marcelina Jureczko, Ewa Felis, Piotr Manczarski

**Affiliations:** 1Institute of Agricultural Engineering, 37a Chełmońskiego Str., Faculty of Life Sciences and Technology, Wroclaw University of Environmental and Life Sciences, 51-630 Wroclaw, Poland; 2Department of Agricultural and Biosystems Engineering, 4350 Elings Hall, Iowa State University, Ames, IA 50011, USA; koziel@iastate.edu; 3Faculty of Agronomy and Bioengineering, 28 Wojska Polskiego Str., Poznan University of Life Sciences, 60-637 Poznań, Poland; jakub.pulka@up.poznan.pl; 4Faculty of Biotechnology and Food Sciences, 37 Chełmońskiego Str., Wroclaw University of Environmental and Life Sciences, 51-630 Wroclaw, Poland; marzena.styczynska@upwr.edu.pl; 5Faculty of Energy and Environmental Engineering, Environmental Biotechnology Department, The Silesian University of Technology, 2 Akademicka Str., 44-100 Gliwice, Poland; joanna.kalka@polsl.pl (J.K.); marcelina.jureczko@polsl.pl (M.J.); ewa.felis@polsl.pl (E.F.); 6Faculty of Building Services, Hydro and Environmental Engineering, Department of Environmental Engineering, 20 Nowowiejska Str., Warsaw University of Technology, 00-653 Warszawa, Poland; piotr.manczarski@pw.edu.pl

**Keywords:** thermal treatment, heavy metals, leachability, toxicity, hydrophobicity, waste to carbon

## Abstract

Improved technologies are needed for sustainable management of sewage sludge (SS). The torrefaction (also known as biomass “roasting”) is considered a pretreatment of SS before use in agriculture. However, it is not known whether the torrefaction has the potential to decrease heavy metals’ (HMs) leachability and the SS toxicity. Thus, the aim of the study was to evaluate the influences of the SS torrefaction parameters (temperature and process time) on HM contents in biochar, HM leachability, and biochar toxicity, and compare them with raw SS. The experiments were designed in 18 combinations (six temperatures, 200, 220, 240, 260, 280, and 300 °C; and three process times—20, 40, 60 min). Standard tests were used to determine HMs content, leachability, and toxicity. Results indicated that the torrefaction did not increase (*p* < 0.05) the HM content in comparison to the raw SS. The leachability of Zn, Ni, Cu, Cr, and Mn from SS biochars was similar to raw SS. However, the degree of leachability varied significantly (*p* < 0.05) from as low as 0.1% for Cu to high as 16.7% for Cd. The leachability of Cd (<16.7%) and Pb (<11.9%) from biochars was higher than from raw SS (<6.1% and <2.4%, respectively). The leachability of Cd from SS biochar, in five torrefaction combinations, was higher than the threshold value for hazardous waste. It is recommended that site-specific decisions are made for torrefaction of SS with respect to its HM content, as the resulting biochar could be considered as hazardous waste, depending on the feedstock. Moreover, the biochar produced under the whole range of temperatures during 20 min retention time significantly (*p* < 0.05) increased the *Daphnia magna* Straus mobility inhibition by up to 100% in comparison to the biochar obtained during 40 and 60 min torrefaction. Taking into account the increased leachability of specific HMs and *D. magna* Straus mobility inhibition, biochar should be considered a potentially hazardous material. Future research should focus on biochar dosage as a fertilizer in relation to its toxicity. Additional research is warranted to focus on the optimization of SS torrefaction process parameters affecting the toxicity.

## 1. Introduction

Biochar is generated from the thermal treatment (torrefaction, pyrolysis) of biowaste [1]. Recent studies indicated that sewage sludge (SS) is a suitable feedstock for torrefaction [2,3,4,5,6]. Torrefaction (also known as “roasting” or “high-temperature drying”) could densify carbon and stabilize SS, thereby offering a sustainable resource recovery solution. The SS torrefaction is a new approach to the management of this biowaste. Since 2013, 121 papers concerning SS torrefaction have been published in Web of Science-indexed journals when searched among via title, abstract, author keywords, and Keywords Plus (keyword search for “torrefaction” and “sewage sludge” in “topic”) (Figure 1). However, only 16 papers included both “torrefaction” and “sewage sludge” in the title. As the SS torrefaction technology is developing, researchers are focusing on the optimization of the process parameters and the improvement of the fuel properties of the biochar [7,8]. Previous studies indicated that both raw and torrefied SS have average fuel properties [4,6].

To date, the main raw SS disposal methods include landfilling, landscaping, incineration, and agricultural applications. Therefore, the agricultural application of torrefied SS has been proposed as an alternative solution for its energetic utilization [6]. However, the aspects of the potential influence of torrefied SS on the environment have not been intensely studied, yet.

Justified concerns about agricultural applications of SS are due to the toxic residues. Raw SS can be hazardous due to residual heavy metals (HMs), persistent organic pollutants, and parasites [9]. In the context of SS and agriculture, it is mainly the HMs that are associated with the highest risk via having adverse effects on the health of humans, animals, and plants, according to their toxicity, accumulation, and biomagnification potential [10].

The SS accumulates as much as 80–90% of HMs from raw wastewater. The typical total HM content ranges from 0.5% to 2% d.m. (dry mass); in some cases, as high as ~4% d.m. [11]. An important factor is HM leachability, which is related to their bioavailability. The leachability of a HM depends on the SS source, pH, nature of the organic matter, oxidation-reduction potential, and electrolytic conductivity. HMs exhibit the most significant leachability in acidic environments when the Fe, Al, and Mn oxides to which they are bound are released. Cd is considered the most leachable, releasing already at 6.5 pH, whereas Cr, Pb, and Hg have the lowest leachabilities. The organic matter in the SS is mainly macromolecular, which results in the formation of stable complexes with HMs. Other forms of organic matter (acids, polysaccharides, amino acids), can, through the complexing ability, contribute to the HMs leachability. The oxidation-reduction potential is associated with oxygenation, moisture exchange, and biological activity of microorganisms. As the potential decreases, the reduction processes decompose organic matter, dissolving the Fe and Mn oxides into the environment [12].

Improved technologies are needed for the sustainable management of SS. The torrefaction has been proposed as a pretreatment of SS before use in agriculture. Hossain et al. [13] torrefied SS and showed that the increased temperature leads to the pH change from acidic to alkaline, which can reduce the leachability of HMs. Soil benefits from biochar application via increased organic matter, improved sorption, improved water retention capacity [14], increased soil microorganisms for organic N mineralization [15], and fertilization [16].

Biochar, a carbon-rich material, has a heterogeneous and highly porous structure, large surface area, and both hydrophilic and hydrophobic character [1]. The hydrophobicity may influence the leachability of HMs from biochars. Research interest in biochar utilization increased during the last decade in the context of biorenewable energy and the need to find applications for its by-products. It is known that biochar properties differ depending on the feedstock and the process parameters, and thus serve different purposes and trigger some unwanted effects [17,18]. The utilization of biochar for agriculture requires practical issues of storage, handling, and application that do not pose any excessive risk to the ecosystem. Therefore, analyzing contaminants in biochar before application to soil is also essential, especially when it is produced from SS containing HMs.

Biochar contains potentially toxic compounds and elements [19], but its toxicity evaluation is inconclusive due to the inherently wide range of materials termed “biochar.” Thus, addressing the gaps in knowledge and finding sustainable management and resource recovery from globally abundant waste (SS) is needed. Specifically, the HM leachability from torrefied SS and the impacts of torrefaction conditions on the toxicity characteristics of biochars produced from the SS could help to make decisions about SS management.

In this research, we explored the feasibility of the thermal pretreatment via torrefaction of SS to address concerns associated with HM content and leachability from the resulting biochars. Torrefaction was chosen for this research due to its lower temperature (and therefore, lower cost) and higher biogenic yield after processing comparing to pyrolysis [4]. The influences of SS torrefaction temperature and (process) residence time on the toxicity of biochar to model organism *Daphnia magna* Straus were also evaluated.

The specific objectives were to (1) evaluate the effects of torrefaction temperatures and the thermal treatment duration on the content of HMs in biochar and compare them with the reference thresholds values for agricultural application, (2) to investigate the influences of the torrefaction temperatures and duration on HMs leachability in comparison with the raw SS, and to (3) test the toxic effects of SS-derived biochars on a model organism. The following working hypothesis was formulated: increases of temperature and duration of the torrefaction increase the HMs content in the produced biochar, decrease the leachability of HMs from the biochar into the environment, and decrease the toxicity of the biochar.

## 2. Materials and Methods

### 2.1. Biochar Production from Sewage Sludge

The SS was acquired from a municipal wastewater treatment plant (WWTP) in Jastrzębie-Zdrój, Poland. This facility is the mechanical-biological WWTP that receives 14,000 m^3^·d^−1^. Wastewater treatment is carried out using methods based on activated sludge in aerobic, anaerobic, and anoxic processes. After separation (in secondary settling tanks) of excess sludge from the treated wastewater, the sludge is subjected to further treatment in fermentation chambers. After the process, digestate sludge is dewatered on a belt filter press. Dewatered sludge samples were collected from a 100 kg initial bulk sample, from which a 10 kg sample was taken and stored in 100 mL vessels at −20 °C. The initial contents of HMs in SS are presented in Table 1.

Biochars were generated in SNOL 8.1/1100 muffle furnace according to the procedure described by [20]. SS was dried for 24 h at 105 °C before the experiment. Heating at a constant rate of 40 °C·min^−1^ always started at ambient temperature and took 5–7 min, depending on the target temperature. The target temperature was maintained for 20, 40, or 60 min, depending on a variant. Then, the samples were left in the furnace to cool down and weighed (with 0.1 mg accuracy) to determine the mass loss. After that, the torrefied SS samples were analyzed for HM content.

CO_2_ was supplied 10 dm^3^∙h^−1^ to ensure inert conditions during the torrefaction process. The heating of the reactor began 5 min after gas was introduced into the reactor. CO_2_ was cut off when the temperature inside the reactor had fallen below 100 °C for safety [21]. A sample crucible could contain between 250 and 300 g of SS for single variant biochar generation. Torrefaction was conducted in 3 retention time variants (20, 40, 60 min), and 6 temperature variants (200, 220, 240, 260, 280, 300 °C). The total number of variants was 36 (two independent factors: temperature and time). Each variant was replicated five times. SS and generated biochar samples were dried and ground in laboratory mills (Retsch SM200, Retsch GmbH, Haan, Germany) to a size below <1 mm before the analyses.

### 2.2. Determination of Physical and Chemical Properties of Sewage Sludge and Produced Biochars

Torrefied and dried (before torrefaction) SS was characterized using standard methods.

Moisture content was measured using the KBC65W (LABFunk, Kluczbork, Poland) laboratory dryer with Radwag PS 3500.R2 (Radwag Wagi Elektroniczne, Radom, Poland) analytical balance following [22].

Heavy metal contents (Zn, Cu, Cr, Mn, Pb, Cd, Ni) were measured by absorption spectroscopy after dry mineralization using Varian Spektr AA 240 FS (Agilent Technologies, Inc, Santa Clara, CA, USA) following [23]. Dry mineralization was carried out with the procedure described below. The homogeneous sample (10 g) was incinerated on the heating plate; then, the samples were mineralized in a muffle furnace for 8 h; the ash was burned for 2 h after dissolving in 2 cm^3^ HNO_3_. The mineralization was transferred quantitatively into 10 cm^3^ vessels using 2 M HNO_3_.

Hydrophobicity was evaluated by measuring the time of water drop penetration (WDPT test). A sample of 5 drops of distilled water at 20 °C was introduced into a sample (shredded up to 1 mm) and distributed on a watch glass with a syringe/pipette. In the case of a longer duration of the penetration process, the tested material was covered with a transparent lid to prevent the rapid evaporation of water droplets. The time of penetration of water by the tested material was measured with a stopwatch and was classified in terms of hydrophobic properties.

### 2.3. Determination of Heavy Metal Leachability from Sewage Sludge and Produced Biochars

The leachability of contaminants and biochars toxicity was determined. Water extracts were obtained according to the following procedure: 30 g each of SS and biochars were weighed with an accuracy of ±0.5 g and transferred quantitatively to 500 mL glass flasks at room temperature (18–27 °C). An amount of distilled water was added to the sample with the 1:9 ratio of biochar:water by mass. After 1 h, the flask was closed and shaken for 4 h to facilitate a complete mixing of the biochar with water. Then the bottle was opened and left for 16 h in static conditions in the dark. The flask was then placed again on a laboratory shaker and shaken for another 4 h. After shaking, the sample was set for 2 h to allow the solids to settle, the clear liquid was decanted, and the rest was centrifuged in a centrifuge at 4500 rpm for 5 min so that the supernatant liquid was clear. In the last step, the supernatant liquid was filtered through paper filters with a basis weight of 84 g m^−2^.

Water extracts were characterized by the analysis described below. HM content in eluates was made using the same process as for SS and biochars. Analyses of NH_4_⁺, NO_3_⁻, NO_2_⁻, P_2_O₅, SO_4_²⁻, and Ntot in water extracts were performed following the [24] standard using the Merck Spectroquant^®^ NOVA 60 Series (Merck KGaA, Darmstadt, Germany). Analysis of the total organic carbon (TOC) content was performed using the Shimadzu TOC-L analyzer (Shimadzu, Kyoto, Japan). Water extracts were used for the determination of the eluates’ toxicities.

### 2.4. Toxicity Bioassay Using Biochar

The *D. magna* Straus test was performed over 2 days of exposure at 20 °C in the dark following the guidelines in [25,26]. Neonates used for the experiment were hatched from commercially available ephippia (Tigret, Warsaw, Poland). Tests were performed in 24-well plates with five crustaceans per well (10 mL of test solution) and four replicates for each undiluted (100%) water extract of biochar obtained according to Section 2.3. This procedure allowed us to measure each whole extract’s toxicity and was appropriate to point out the differences in the toxicity of extracts obtained in different variants (temperature/time) of torrefaction. The medium in the blind sample was aerated tap water. The test endpoint was the inhibition of mobility.

### 2.5. Statistical Analyses

The detailed ANOVA evaluation of differences between mean values in relation to temperature and torrefaction residence time was performed with the application of the post-hock Tukey test at the *p* < 0.05 significance level. Linear regressions at *p* < 0.05 significance level between pollutants’ concentrations in water extracts, pH, and toxicological parameters were estimated. MS Office Excel (Microsoft Corp., Redmond, WA, USA) and Statistica 12 (StatSoft, Inc., TIBCO Software Inc. Palo Alto, CA, USA) software were used to conduct statistical analysis.

Data presenting the fuel properties of SS and biochars (lost on the ignition, ash content, C, H, N, O, higher heating value, lower heating values) and fertilizer properties of SS and biochars (Ca, Mg, K, Na content) obtained in this experiment were published by [6]. Raw data of solid content, mass yield, HM contents in SS and biochars, hydrophobicity, toxicity, and HMs’ and other pollutants’ concentrations in water extracts from SS and biochars are included as Appendix A.

## 3. Results and Discussion

### 3.1. The Leachability of Biochars Obtained from Sewage Sludge via Torrefaction

The hydrophobicity test was used to determine the leachability of HMs from biochars obtained from SS via torrefaction. As the temperature and duration of the torrefaction increased, the hydrophobicity of the resulting biochar increased (Table 2) from “moderate” to the “extreme.” The only material classified as “slightly” hydrophobic was raw SS. In the case of torrefied SS obtained from 20 min torrefaction, the degree of hydrophobicity ranged from “moderate” to “extreme” for materials generated at 200 and 300 °C, respectively (Table 2). This trend is related to the decrease in O/C and H/C molar ratios, resulting from the increases in the concentrations of long-chain and aromatic compounds that are less susceptible to degassing compared to the other carbohydrates found in the SS—hence the reduction of their polarity and their reduction of water affinity [6].

Polarity is generally known to increase the hydrophobicity in biochars [27]. A similar confirmation of the increase in hydrophobicity, along with the rise of the torrefaction temperature, was also documented for biochars produced from other feedstocks [28,29]. The hydrophobicity of the biochar applied to the soil may decline over time due to the biochar’s reaction with compounds contained in the soil and microorganisms. However, slow reduction of hydrophobicity over time, combined with the high content of nutrients in the biochar, is desirable; i.e., biochars have the potential to become, in a sense, long-term reservoirs of nutrients.

### 3.2. The Influences of Temperature and Time of the Torrefaction on Heavy Metal Content in Biochars from Sewage Sludge

In general, torrefaction did not significantly increase (*p* < 0.05) the HM content in comparison to the raw SS. In addition, the content of as many as seven HMs (Cu, Mn, Cd, Pb, Zn, Cr, Ni) in biochars from SS (out of seven total targeted in the analyses) met the requirements to be below the maximum concentrations allowed for the use of municipal SS in agriculture and land reclamation for agricultural purposes.

The torrefaction of SS did not show a statistically significant (*p* < 0.05) effect of temperature and retention time on the Cu content in the biochar samples (Figure 2, Figure 3 and Figure 4). The Cu content in the raw SS amounted to 33.5 mg Cu·kg^−1^ d.m. (Table 1). The mean Cu content in biochars ranged from 29.09 to 41.48 mg Cu·kg^−1^ d.m. for all temperatures and 20 min (Figure 2). In biochars generated for 40 min, the mean Cu content ranged from 25.7–49.2 mg Cu·kg^−1^ d.m. (Figure 3). Extending the torrefaction to 60 min did not change significantly (*p* < 0.05) the mean Cu content in biochars, which ranged from 31.7–48.5 mg Cu·kg^−1^ d.m. (Figure 4). The Cu content in biochars from SS met the requirements of maximum concentrations (1000 mg Cu·kg^−1^ d.m.) for the use of municipal SS in agriculture and land reclamation for agricultural purposes [30].

The results for Cu differ from those obtained in the work of [31], wherein torrefaction at 300 °C caused significant increases in the Cu contents of three different SSs: 611–1034, 401–480, and 451–687 mg Cu·kg^−1^ d.m. The results cited above show significant variability in this respect. Similarly, Hossain et al. [13] showed an increase of Cu content in biochar in comparison to raw SS from 810 to 1150 mg Cu·kg^−1^ d.m. The lack of statistically significant (*p* < 0.05) differences between the raw SS and the biochar in this work could have been caused by the relatively low concentration of Cu in the raw SS (an order of magnitude lower than in case of Hossain et al., [13] and Lu et al., [31] and high variability of Cu content in raw SS (18.5%, Table 1).

The Mn content is not considered as a pollutant per threshold value for SS and biochar application to soil. However, it is one of the micronutrients essential to plant growth. The Mn content also showed no statistically significant (*p* < 0.05) influence (Figure 2, Figure 3 and Figure 4) of torrefaction temperature and time. The Mn content for raw SS was 120.6 mg Mn·kg^−1^ d.m. (Table 1). Mn content values fluctuated from 117.2 to 136.6 mg Mn·kg^−1^ d.m. (Figure 2), from 104.5 to 136.4 mg Mn·kg^−1^ d.m. (Figure 3), and from 103.2 to 141.5 mg Mn·kg^−1^ d.m. (Figure 4), in biochars generated during 20, 40, and 60 min of retention time, respectively.

The Cd content in biochars increased with the duration of the process (Figure 2, Figure 3 and Figure 4). However, Cd content in the biochar was not statistically (*p* < 0.05) different from the content in the raw SS (8.1 mg Cd·kg^−1^ d.m., Table 1). The mean Cd contents were 7.6–12.3 mg Cd·kg^−1^ d.m. (Figure 2), 5.9–9.5 mg Cd·kg^−1^ d.m. (Figure 3), and 4.4–9.6 mg Cd·kg^−1^ d.m. (Figure 4), in biochars generated during 20, 40, and 60 min of retention time, respectively. The Cd contents in biochars presented by Lu et al. [31] showed statistically significant increases for three types of SS: 3.39–5.68 mg Cd·kg^−1^ d.m, 2.28–3.30 mg Cd·kg^−1^ d.m, and 5.26–7.45 mg Cd·kg^−1^ d.m. A similar (relatively small) increase in Cd content was reported by Hossain et al. [13], i.e., from 2.07 to 2.62 mg Cd·kg^−1^ d.m. Similarly, as in the case of Cu and Mn contents, the reason for the lack of the statistically significant (*p* < 0.05) differences in this research could be the high variation coefficient of Cd content in the raw SS (16.25%, Table 1). Additionally, high standard deviations from the mean values were noted in all resulting biochars (Figure 2, Figure 3 and Figure 4). Obtained data for Cd content in the present experiment meet the maximal concentration (20 mg Cd·kg^−1^ d.m.) for the use of municipal SS in agriculture and the reclamation of land for agricultural purposes [30]. However, the threshold values of Cd content for “organic fertilizers” (5 mg Cd·kg^−1^ d.m.) have been exceeded [32], and therefore, exclude the produced biochars from this type of utilization.

The consequence for this particular SS source is that torrefaction should not be considered as waste recycling (according to EU waste directive 2008/98/EC [33]), as the biochars would still have a status of “waste.” Moreover, the torrefaction changed the classification of raw SS from “190805—sludges from treatment of urban wastewater” to biochar considered as “190118—pyrolysis wastes other than those mentioned in 19 01 17 (Commission Decision 2000/532/EC) [34].” Thus, clearly, site-specific solutions are needed, and it is recommended to consider a more comprehensive legal evaluation of WWTP-specific feedstock of SS and resulting biochars.

The Pb content in the raw SS was 72.0 mg Pb·kg^−1^ d.m. (Table 1). The Pb content in biochars did not show any statistically significant (*p* < 0.05) influence from the temperature and torrefaction retention time (Figure 2, Figure 3 and Figure 4). The mean content of Pb ranged from 54.0 to 169 mg Pb·kg^−1^ d.m. (Figure 2), 53.2 to 128 mg Pb·kg^−1^ d.m. (Figure 3), and 28.5 to 71. 9 mg Pb·kg^−1^ d.m. (Figure 4), in biochars generated during 20, 40, and 60 min of retention time, respectively. Other authors reported the increase of Pb content in biochars with temperature. Hossain et al. [13] observed that the Pb content increased from 86.5 to 115 mg Pb·kg^−1^ d.m. Comparable results were reported by Mierzwa-Hersztek et al. [35] for biochar obtained in 20 min and 300 °C. Additionally, Lu et al. [32] reported Pb contents of three different sludge samples processed at 300 °C significantly increased from 136 to 242 mg Pb·kg^−1^, 138 to 189 mg Pb·kg^−1^ d.m., and 224 to 350 mg Pb·kg^−1^ d.m, respectively. The obtained biochars meet the requirements of maximum concentrations (750 mg Pb·kg^−1^ d.m.) for the use of municipal SS in agriculture and the reclamation of land for agricultural purposes [30]. However, in the case of torrefaction temperature 300 °C, the threshold values of Pb content for “organic fertilizers” (140 mg Cd·kg^−1^ d.m.) were exceeded [32]; that would exclude this biochar from this type of reuse.

The mean Zn content in raw SS was 212 mg Zn·kg^−1^ d.m. (Table 1). Zn content in biochars increased with the process temperature (Figure 2, Figure 3 and Figure 4). The mean Zn contents in biochars ranged from 222 to 279 mg Zn·kg^−1^ d.m. (Figure 2), 174 to 348 mg Zn·kg^−1^ d.m. (Figure 3), and 206 to 266 mg Zn·kg^−1^ d.m. (Figure 4), in biochars generated during 20, 40, and 60 min of retention time, respectively. The Zn content in biochar was not statistically different (*p* < 0.05) from that in raw SS, except for the case of the biochar produced over 40 min at 260 °C (i.e., 348 mg Zn·kg^−1^ d.m.) (Figure 3; Table 1). A similar trend was reported by Hossain et al. [13] and Lu et al. [31], wherein the Zn content increased from 1350 to 1675 mg Zn·kg^−1^ d.m., and 1240 to 1910 mg Zn·kg^−1^ d.m. respectively, for SS torrefaction at 300 °C. The present results meet the requirements of maximum concentrations (2500 mg Zn·kg^−1^ d.m.) for the use of municipal SS in agriculture and land reclamation for agricultural purposes [30]—Zn is not considered as a pollutant in organic fertilizers [32].

The mean Cr content in raw SS was 3.6 mg Cr·kg^−1^ d.m. (Table 1). The Cr content did not significantly change with the increase of the temperature and torrefaction retention time (Figure 2, Figure 3 and Figure 4). The mean content of Cr in biochars ranged from 1.96 to 8.24 mg Cr·kg^−1^ d.m. (Figure 2), 1.84 to 4.10 mg Cr·kg^−1^ d.m. (Figure 3), and 1.09 to 2.59 mg Cr·kg^−1^ d.m. (Figure 4), in biochars generated during 20, 40, and 60 min of retention time, respectively. The observed decrease in Cr may be associated with partial fractionation and degassing, increased retention time, the exposition of material to increased heat changes, and more robust decomposition. However, this process is more common for pyrolysis [36]. The obtained biochars also meet the requirements for the use of municipal SS in agriculture and land reclamation for agricultural purposes (500 mg Cr·kg^−1^ d.m. [30]) and (100 mg Cr·kg^−1^ d.m.) presented in [32] for “organic fertilizers”.

The Ni content in biochars decreased with the duration of the process. However, the Ni content in biochars was not statistically (*p* < 0.05) different in relation to the raw SS. The mean content of Ni in biochars ranged from 10.71 to 15.01 mg Ni·kg^−1^ d.m. (Figure 2), 10.69 to 15.36 mg Ni·kg^−1^ d.m. (Figure 3), and 7.77 to 13.04 mg Zn·kg^−1^ d.m. (Figure 4), in biochars generated during 20, 40, and 60 min of retention time, respectively. The results obtained in the torrefaction of SS by Hossain et al. [13] and Lu et al. [31] showed higher Ni content in biochars in relation to the raw SS. The Ni content in biochars meets the requirements for the use of municipal SS in agriculture and land reclamation purposes [30,32] (100 and 60 mg Ni·kg^−1^ d.m.).

In summary, the HM contents in biochars were below the threshold values for the application of SS [30] for agriculture. However, in the case of Cd (all biochars) and Pb (one biochar), the threshold values of the mentioned HMs in organic fertilizers [32] were exceeded. That means that biochars produced from SS would still be considered as “waste” (EU code 190118), which requires special treatment according to Directive 2008/98/EC [33], defined as recovery operation R10 Land treatment resulting in benefit to agriculture or ecological improvement. According to another Polish regulation, the waste classified as “190118” is not allowed to be recovered in the R10 process [37], but raw SS is permitted (under special restrictions). In the present case, due to the high content of Cd in the SS, the torrefaction of this type of SS creates an additional challenge. This is because the fuel properties of biochars are not attractive enough to consider the SS biochar as a solid fuel [6], and the high HM contents limit its agricultural use.

The initial HM level of contamination in raw SS is crucial for decision making concerning the application of torrefaction. It is thus recommended that a comprehensive assessment of the local (at least country/state level) regulations pertaining to the HM content in the SS feedstock is site-specific and may also vary with the WWTPs size and wastewater-type treated.

In addition, the results presented are associated with a high degree of variability in HM content. In most cases, the temperature and torrefaction duration did not influence significantly (*p* < 0.05) the contents of the HMs in biochars. In a few instances, the contents of HMs in biochars were lower than in raw SS. That could have been due to the very high heterogeneity of the material used—very high variation coefficients (Table 1) and heterogeneity of the properties of biochars; high values of standard deviations (Figure 2, Figure 3 and Figure 4). It should be noted that the lack of statistically significant differences between the results obtained for biochar and SS in the vast majority of cases reported here should be thoroughly verified in future research. That may have resulted from considerable standard deviations obtained during the analysis, which may be related to the analysis of just five separately generated samples. The required number of samples (n) for each variant could be determined, according to [38]:(1)$$n=\left({\frac{t_{\left(\alpha ;n-1\right)}\cdot \mathrm{var}\;\mathrm{coeff}\left(x\_i\right))}{\varepsilon }}^{2}\right)$$
where *t*_(α;n−1)_ is a demand on the confidence level (expressed by the confidence coefficient (z-value)) of the t-distribution, the variance of the population (expressed by the coefficient of variation: var coeff (xi)); ε is the desired accuracy of the results (expressed by the maximum allowance for random sampling error).

Considering the determined standard deviations desired, 10% accuracy, the confidence level (0.05) threshold, and the SWA-Tool [38] methodology, the recommended number of repetitions of the same variant is as high as 19. Thus, the results reported (five repetitions) should be treated as preliminary research conducted on one type of SS; therefore, further investigations are required.

### 3.3. Heavy Metal Leachability from Biochars in Relation to Torrefaction Temperature and Retention Time

The low leachability of Zn, Ni, Cu, Cr, and Mn from SS biochars was confirmed. The degree of leachability varied significantly (*p* < 0.05) from as low as 0.1% for Cu to 16% for Cd. The mobility of Cd and Pb from biochars was higher (and lower in the case of Zn) than from raw SS. However, the pH of the aqueous extracts could influence the mobility of HMs. The pH range of generated eluates from SS biochar was from 6.92 to 7.47 (Table 3) and had a statistically significant (*p* < 0.05) decrease as the temperature and duration of the process increased. The increase in pH was significantly lower than that reported by Jin et al. [39], but it has to be pointed out that in the case of Jin et al.’s research, the process temperature was much higher (400 and 600 °C). The increase in pH value in the present experiment was similar to the result shown by Mierzwa-Hersztek et al. [35]. The neutral pH range is desirable, as the mobility of HMs tends to be low [40].

The influence of the torrefaction on the mobility of HMs was confirmed by their low leachability. The Zn leachability was higher for SS eluates compared to biochars, an opposite trend to the other target HMs. The percentage of HM leaching in relation to their content in the torrefied SS was, on average, from 0.1% for Cu to 16% for Cd (Table 3). Such low levels of HMs (with the exception of Cd) are typical for biochars and other products of thermal treatment [41,42,43]. The Cr and Ni concentrations were below the detection level (0.01 mg·kg^−1^ d.m.) in all eluates. The Cu concentrations were characterized by maximal leachability of 0.7% (Table 3) and were only detected in three variants from 18 in total.

**Table 3 materials-13-03544-t003:** Mean values of heavy metal leachability and pH from water extracts of raw sewage sludge and torrefied SS obtained from torrefaction of sewage sludge in six temperature variants, with three retention times; (−) indicates the value was below the detection level of 0.01 mg·kg^−1^ d.m.

Times, Min	Temperature, °C	Mass Yield, %	pH	Cu	Mn	Cd	Pb	Zn	Cr	Ni	Cu	Mn	Cd	Pb	Zn	Cr	Ni
				mg·kg^−1^ d.m.							leachability						
Sewage Sludge			7.2	−	0.93	0.49	1.73	2.68	−	−	−	0.8%	6.1%	2.4%	1.3%	−	−
20	200	94	7.4	0.01	0.84	0.53	2.08	1.11	−	−	−	0.7%	7.0%	2.4%	0.5%	−	−
20	220	94	7.3	0.23	0.81	0.54	2.60	0.87	−	−	0.7%	0.7%	5.6%	3.5%	0.4%	−	−
20	240	94	7.2	−	0.87	0.46	1.59	0.43	−	−	−	0.7%	4.1%	2.4%	0.2%	−	−
20	260	93	7.5	−	0.75	0.47	1.54	1.46	−	−	−	0.6%	6.2%	2.9%	0.5%	−	−
20	280	91	7.3	−	1.76	1.09	3.53	1.26	−	−	−	1.3%	12.2%	6.2%	0.5%	−	−
20	300	89	7.1	−	1.07	0.78	2.83	0.83	−	−	−	0.8%	6.4%	1.7%	0.3%	−	−
40	200	91	7.1	−	1.27	0.73	2.27	0.72	−	−	−	1.2%	8.6%	4.3%	0.4%	−	−
40	220	92	7.1	−	1.53	1.04	3.26	1.15	−	−	−	1.3%	16.7%	3.3%	0.5%	−	−
40	240	90	7.1	−	1.30	0.83	4.34	0.91	−	−	−	1.2%	14.0%	7.1%	0.5%	−	−
40	260	90	7.0	−	1.20	1.22	7.77	1.79	−	−	−	0.9%	16.0%	11.9%	0.5%	−	−
40	280	88	7.1	−	2.08	0.85	3.87	0.91	−	−	−	1.7%	8.9%	3.0%	0.3%	−	−
40	300	83	7.0	−	1.33	1.06	4.67	1.40	−	−	−	1.0%	13.8%	7.4%	0.5%	−	−
60	200	90	7.1	0.04	1.36	1.06	4.24	1.18	−	−	0.1%	1.3%	15.9%	7.6%	0.5%	−	−
60	220	91	7.1	−	1.07	0.83	3.09	0.61	−	−	−	0.9%	8.6%	4.3%	0.3%	−	−
60	240	89	7.1	−	1.09	0.67	2.52	0.66	−	−	−	0.8%	11.3%	5.9%	0.3%	−	−
60	260	88	7.1	0.05	0.94	0.57	2.49	0.59	−	−	0.2%	0.9%	13.0%	8.7%	0.3%	−	−
60	280	87	6.9	−	1.23	0.87	3.70	0.71	−	−	−	0.9%	15.1%	7.9%	0.3%	−	−
60	300	81	7.0	−	1.45	0.92	4.25	1.16	−	−	−	1.2%	15.8%	11.7%	0.5%	−	−
Hazardous waste threshold value *		−	−	50	−	1	10	50	10	10	−	−	−	−	−	−	−

* [44]; shadow: over-limit values.

Overall, the HMs’ leachability from biochars should be considered low. Some exceptions are associated with Pb (two samples with leachability > 10%) and Cd (10 samples with leachability > 10%). Additionally, the leachability of Pb ranged from 1.7% to 11.9% and for Cd from 4.1% to 16.7%, respectively. The highest leachability of Cd and Pb were observed for biochars generated at 40 and 60 min torrefaction. Higher leachabilities of Cd and Pb in comparison to Jin et al. [39] were likely associated with the lower pH of the eluate compared to the cited work.

The most problematic HM in the present study was Cd. In the Polish environmental legislative system, hazardous waste is classified by the waste producer; however, the pollutant leaching tests must be completed first to determine if landfilling is permissible. Three types of landfills are defined depending on the waste type: neutral, hazardous, and non-neutral and non-hazardous. For each type of waste, the threshold values of pollutant leachability have been defined [44]. In the case of Cd, the threshold value for hazardous waste is 1 mg·kg^−1^ d.m. If the leachability is within the range between 1 and 5 mg·kg^−1^ d.m., the waste should be considered as hazardous waste, and it must be landfilled only at the hazardous waste landfill site. In the present study, in the cases of five variants (per 18), the threshold value for Cd was exceeded and ranged from 1.06 to 1.22 mg·kg^−1^ d.m. (Table 3). In this case, torrefaction increased the HMs leachability (about doubled compared with the raw SS) and generated hazardous waste. Therefore, the initial leachability of HMs from raw material should be completed before deciding on SS torrefaction.

The contents of individual trace elements in SS may vary depending on the source [45]. Industries that can significantly increase the HM content in SS include metal and metalworking, tannery, chemical, and pulp & paper. More research is needed on the influences of different SS types on the content and the leachability of HMs.

### 3.4. SS Biochars’ Ecotoxicity in Relation to Torrefaction Temperature and Process Retention Time

The question on the influence of SS transformation due to torrefaction on the produced biochar toxicity level has been investigated here. Confirmation of raw SS toxicity can be found in the previous works [46,47,48].

Toxicity analysis of *D. magna* Straus performed on the undiluted water extracts showed the highest mobility inhibitions for samples of dry SS and torrefied SS (for 20 min). The obtained results indicate 100% mobility inhibition of organisms exposed to the dry SS eluate. A similar trend was visible in the case of water extracts from the biochars obtained between 200 and 260 °C, where already on the first day of exposure, the immobilization reached 100%. The first observed differences in mobility inhibition were for biochars generated at 280 and 300 °C, for which the average inhibitions were 69% and 89%, respectively (Figure 5). For comparison, the highest mobility inhibition obtained for samples generated at 40 and 60 min was 33%. The lowest inhibition was measured for eluates from biochars generated at 40 min between 240 and 280 °C. Toxicity decreased towards *D. magna* with increasing temperature, and the process retention time was also confirmed by Zielińska and Oleszczuk [36], who analyzed the impact of SS pyrolysis on toxicity.

One reason for higher toxicity in the above conditions may be due to higher ammonia leaching (Table 4). The most significant correlation between potential contaminants concentration and mobility inhibition was determined for nitrogen and ammonia (Table 5). These results were consistent with the previous work [49,50], where the toxicity towards *D. magna* was linked to ammonia and chemical oxygen demand (COD) concentrations. A strong correlation was also apparent for pH levels and total P (Table 5). In this experiment, the influences of only dissolved inorganic compounds were tested. Three groups of potentially toxic substances, namely, metals and metalloids (e.g., As, Cu, Pb, Ni, Zn, etc.); polycyclic aromatic hydrocarbons; and dioxins, are most likely to be present in biochar [51,52]. Therefore, the testing of toxicity could also be performed for the presence of selected organic pollutants (e.g., volatile organic compounds) for a broader evaluation of temperature and retention time’s impact on biochar toxicity. Further studies in this field are required.

Apart from the potential risk factor of utilizing biochar as a fertilizer, another important issue is the risk of biochar production itself. It is vital to conduct the life cycle assessment, calculate the carbon footprint, and narrow down the specific influences of biochar production and utilization on the climate change [53,54]

## 4. Conclusions

The executed experiment yielded the following observations:The torrefaction process did not increase the contents of HMs in the biochars in relation to the raw SS. The individual HM concentrations in the biochars were not significantly dependent on the process temperature and retention time. The HM content in the biochar was relatively low and fulfilled the requirements of the Polish Regulation for agricultural use of SS. However, because of the high content of Cd from 4.4 to 12.3 mg Cd·kg^−1^ d.m. (also present in raw SS), it excluded the biochars from agricultural use as an “organic fertilizer.”Consistently decreased leachability of HMs from biochar in comparison to raw SS was measured only in the case of Zn. For other metals, no such trend was visible, i.e., no significant trend for specific variants compared to raw SS.The leachability of Cd (12%) from torrefied SS increased compared to the untreated SS (6%), and exceeded the threshold values for hazardous waste.In the case of crustaceans *D. magna* Straus, the toxicity of water extracts from biochars was reduced (compared to the raw SS) as the temperature and residence time of torrefaction increased.

The results emphasize a very practical significance of feedstock quality (especially the HM content) if SS recycling to organic fertilizer via torrefaction is considered. Therefore, it is extremely important that the raw SS is analyzed, and laboratory-scale trials with biochar production from SS are conducted before the decision to scale-up the SS torrefaction is made. There is the risk that for some cases (i.e., SS with initially high HM content, as in the analyzed case) hazardous waste will be produced. Additional analyses should be considered in the future, wherein the simulation of HMs’ fates after the SS biochar application to the soil under lysimetric conditions should be investigated. The obtained results also confirmed the hypothesis that the toxicity of the obtained biochar is reduced with the increase in the torrefaction temperature and retention time of the SS in the torrefaction reactor. However, broadening the spectrum of tested model organisms should be considered. Future research should consider not only on toxicity, and possible hazardous compound composition and leachability, but also the broader influence of this carbon sequestration on the carbon footprint and climate change as a whole.

## Figures and Tables

**Figure 1 materials-13-03544-f001:**
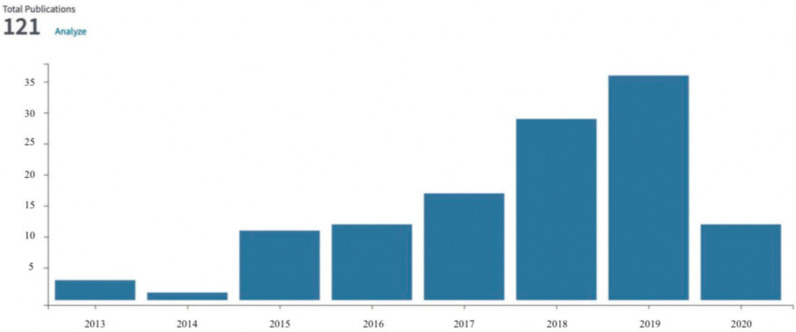
The number of papers found in the Web of Science database.

**Figure 2 materials-13-03544-f002:**
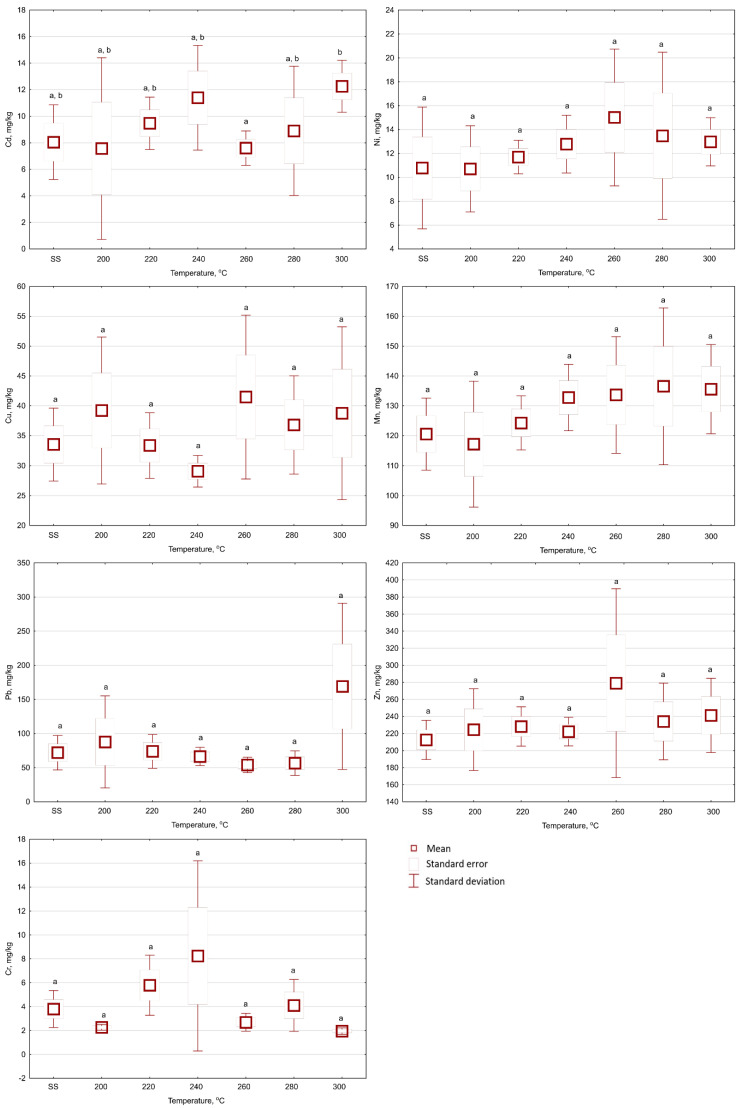
The influences of temperature on Cd, Ni, Cu, Mn, Pb, Zn, and Cr contents in biochars produced with 20 min of retention time under 200–300 °C temperatures (SS—raw sewage sludge). Letters indicate the statistically significant (*p* < 0.05) differences between mean values.

**Figure 3 materials-13-03544-f003:**
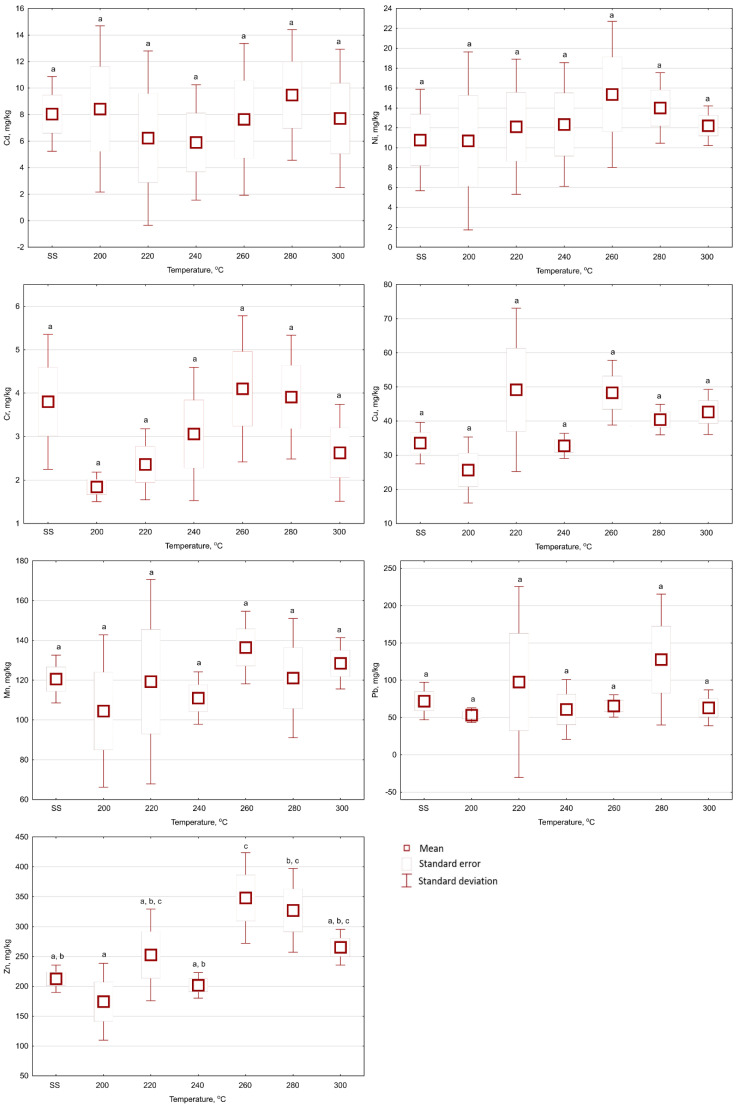
The influences of temperature on Cd, Ni, Cu, Mn, Pb, Zn, and Cr contents in biochars produced with 40 min of retention time under 200–300 °C temperatures (SS—raw sewage sludge). Letters indicate the statistically significant (*p* < 0.05) differences between mean values.

**Figure 4 materials-13-03544-f004:**
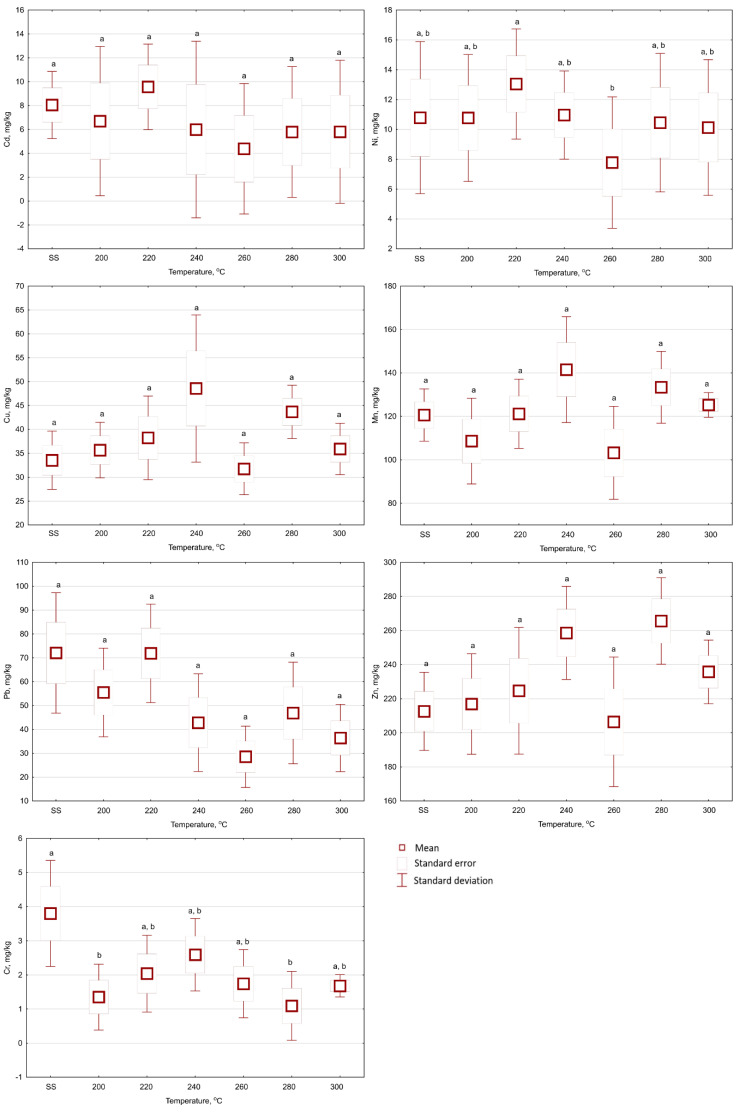
The influences of temperature on Cd, Ni, Cu, Mn, Pb, Zn, and Cr contents in biochars produced with 60 min of retention time under 200–300 °C temperatures (SS—raw sewage sludge). Letters indicate the statistically significant (*p* < 0.05) differences between mean values.

**Figure 5 materials-13-03544-f005:**
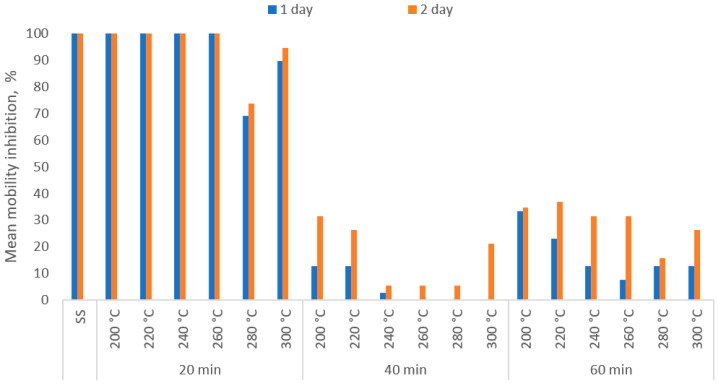
The average mobility inhibition of *D. magna* Straus during 2 days of exposure to undiluted water extract from sewage sludge and torrefied SS obtained from torrefaction of sewage sludge in 6 temperature variants (200–300 °C; SS—raw sewage sludge) with three retention times (20–60 min).

**Table 1 materials-13-03544-t001:** Average raw sewage sludge properties (±SD, N = 5).

The Property	Values ± SD	Variation Coefficient, %
dry mass (d.m.), %	20.3 ± 0.3	1.48
Zn, mg·kg^−1^, d.m.	212.6 ± 23.4	11.00
Cu, mg·kg^−1^, d.m.	33.5 ± 6.2	18.51
Cr, mg·kg^−1^, d.m.	3.8 ± 1.6	42.11
Mn, mg·kg^−1^, d.m.	120.6 ± 12.3	10.20
Pb, mg·kg^−1^, d.m.	72.0 ± 25.8	35.42
Cd, mg·kg^−1^, d.m.	8.0 ± 1.3	16.25
Ni, mg·kg^−1^, d.m.	10.8 ± 2.3	21.30

**Table 2 materials-13-03544-t002:** Degrees of hydrophobicity of dry sludge and torrefied sewage sludge (SS) obtained from torrefaction of sewage sludge in six temperature variants, with three residence times.

Process Time, Min	Temperature, °C	WDPT ± SD, s	Hydrophobicity Level
Dry sludge		83 ± 14	Slightly hydrophobic
20	200	222 ± 24	Moderately hydrophobic
	220	340 ± 28	Hydrophobic
	240	499 ± 28	Hydrophobic
	260	559 ± 43	Hydrophobic
	280	745 ± 48	Strongly hydrophobic
	300	<3600	Extremely hydrophobic
40	200	340 ± 25	Hydrophobic
	220	434 ± 35	Hydrophobic
	240	535 ± 32	Hydrophobic
	260	1482 ± 240	Severely hydrophobic
	280	<3600	Extremely hydrophobic
	300	<3600	Extremely hydrophobic
60	200	454 ± 47	Hydrophobic
	220	750 ± 37	Strongly hydrophobic
	240	2520 ± 183	Severely hydrophobic
	260	<3600	Extremely hydrophobic
	280	<3600	Extremely hydrophobic
	300	<3600	Extremely hydrophobic

**Table 4 materials-13-03544-t004:** Mean values of contaminants’ concentrations and pH values from water extracts of raw sewage sludge and torrefied SS obtained from torrefaction of sewage sludge in 6 temperature variants, with three retention times.

Time. Min	Temperature. °C	TOC	Ntot	N-NH_4_	N-NO_2_	N-NO_3_	SO_4_	Cl	Ptot (P-PO_4_)	PO_4_-P	pH	Mn	Cd	Pb	Zn	Cu	Cr	Ni
		mg·dm^−3^										mg·dm^−3^						
Sewage Sludge		24,040.8	6660.0	2341.8	5.8	75.6	7254.0	1998.0	1378.8	741.6	7.2	0.97	0.51	1.80	2.78	-	-	-
20	200	17,379.0	5526.0	1904.4	5.2	61.2	8784.0	1794.6	1000.8	759.6	7.4	0.87	0.55	2.13	1.14	0.01	-	-
20	220	14,275.8	5508.0	2134.8	5.4	88.2	6300.0	1992.6	986.4	725.4	7.3	0.84	0.55	2.69	0.90	0.24	-	-
20	240	16,250.4	4752.0	1594.8	6.5	95.4	7218.0	1834.2	865.8	469.8	7.2	0.90	0.48	1.64	0.44	-	-	-
20	260	17,226.0	4158.0	943.2	5.4	61.2	12,258.0	1116.0	860.4	457.2	7.5	0.77	0.48	1.58	1.50	-	-	-
20	280	18,630.0	4716.0	1099.8	6.1	64.8	16,470.0	988.2	995.4	495.0	7.3	1.80	1.11	3.60	1.29	-	-	-
20	300	16,273.8	4284.0	799.2	4.9	59.4	15,822.0	1463.4	792.0	478.8	7.1	1.08	0.79	2.87	0.85	-	-	-
40	200	18,410.4	3798.0	1530.0	5.2	55.8	17,010.0	1261.8	777.6	529.2	7.1	1.32	0.76	2.37	0.75	-	-	-
40	220	20,435.4	3150.0	1297.8	5.2	61.2	12,654.0	1627.2	687.6	543.6	7.1	1.55	1.05	3.30	1.16	-	-	-
40	240	19,184.4	3168.0	826.2	4.3	73.8	12,330.0	1265.4	646.2	486.0	7.1	1.33	0.84	4.42	0.92	-	-	-
40	260	18,930.6	3636.0	579.6	4.9	70.2	11,988.0	1366.2	822.6	550.8	7.0	1.24	1.27	8.07	1.86	-	-	-
40	280	9686.7	1548.0	367.2	2.4	55.8	8478.0	453.6	394.2	302.4	7.1	2.13	0.87	3.96	0.93	-	-	-
40	300	14,715.0	1674.0	284.4	2.2	45.0	8730.0	309.6	482.4	356.4	7.0	1.36	1.09	4.79	1.44	-	-	-
60	200	20,901.6	5148.0	982.8	8.8	111.6	9936.0	1279.8	444.6	394.2	7.1	1.40	1.09	4.38	1.22	0.04	-	-
60	220	21,378.6	3798.0	687.6	5.4	84.6	11,592.0	1252.8	574.2	369.0	7.1	1.11	0.85	3.20	0.63	-	-	-
60	240	15,427.8	3024.0	421.2	8.6	95.4	10,962.0	1323.0	415.8	221.4	7.1	1.12	0.69	2.59	0.68	-	-	-
60	260	14,554.8	2574.0	430.2	7.4	90.0	11,304.0	1476.0	401.4	207.0	7.1	0.96	0.58	2.55	0.60	0.05	-	-
60	280	12,049.9	1260.0	450.0	8.8	100.8	7992.0	1548.0	464.4	181.8	6.9	1.26	0.89	3.80	0.72	-	-	-
60	300	12,398.4	1854.0	655.2	4.9	86.4	8676.0	1807.2	379.8	194.4	7.0	1.48	0.94	4.35	1.18	-	-	-

**Table 5 materials-13-03544-t005:** Correlation results and function parameters among contaminant concentrations, pH, and mobility inhibitions for *D. magna* Straus. Statistical significance (*p* < 0.05) is marked by the red font.

**1st Day Mobility Inhibitions**			
**Pollutant**	**Function Parameters**	***p***	**r^2^**
TOC, mg·dm^−3^	y = 16140.9314 + 19.5944·x	0.3373	0.0542
N_tot_, mg·dm^−3^	y = 2538.7216 + 27.8626·x	0.00006	0.6226
N-NH_4_, mg·dm^−3^	y = 575.3784 + 10.6358·x	0.0004	0.5278
N-NO_2_, mg·dm^−3^	y = 5.4842 + 0.004·x	0.7049	0.0086
N-NO_3_, mg·dm^−3^	y = 75.6653 − 0.0016·x	0.9882	0.0000
SO_4_, mg·dm^−3^	y = 11288.2594 − 11.0422·x	0.5426	0.0222
Cl, mg·dm^−3^	y = 1160.6702 + 5.1987·x	0.0344	0.2373
P_tot_ (P-PO_4_), mg·dm^−3^	y = 497.8575 + 4.9533·x	0.0001	0.5889
PO_4_-P, mg·dm^−3^	y = 334.7252 + 2.6644·x	0.0036	0.4019
pH	y = 7.0289 + 0.0027·x	0.00004	0.6404
Mn, mg·dm^−3^	y = 1.4295 − 0.0046·x	0.0110	0.3240
Cd, mg·dm^−3^	y = 0.9612 − 0.0036·x	0.0037	0.3993
Pb, mg·dm^−3^	y = 4.2830 − 0.0219·x	0.0049	0.3805
Zn, mg·dm^−3^	y = 0.9938 + 0.0027·x	0.3808	0.0455
**2nd Day Mobility Inhibitions**			
	**Function parameters**	***p***	**r^2^**
TOC, mg·dm^−3^	y = 15891.8326 + 21.4943·x	0.3509	0.0513
N_tot_, mg·dm^−3^	y = 2180.3293 + 30.6487·x	0.0001	0.5927
N-NH_4_, mg·dm^−3^	y = 436.2804 + 11.7456·x	0.0006	0.5064
N-NO_2_, mg·dm^−3^	y = 5.4323 + 0.0044·x	0.7111	0.0083
N-NO_3_, mg·dm^−3^	y = 76.4184 − 0.0165·x	0.8898	0.0012
SO_4_, mg·dm^−3^	y = 11272.3512 − 8.9539·x	0.6623	0.0115
Cl, mg·dm^−3^	y = 1092.2603 + 5.7496·x	0.0385	0.2283
P_tot_ (P-PO_4_), mg·dm^−3^	y = 436.1408 + 5.4082·x	0.0003	0.5523
PO_4_-P, mg·dm^−3^	y = 304.7134 + 2.8447·x	0.0066	0.3604
pH	y = 6.9924 + 0.003·x	0.00005	0.6286
Mn, mg·dm^−3^	y = 1.5122 − 0.0056·x	0.0059	0.3676
Cd, mg·dm^−3^	y = 1.0236 − 0.0043·x	0.0018	0.4443
Pb, mg·dm^−3^	y = 4.7142 − 0.0271·x	0.0014	0.4590
Zn, mg·dm^−3^	y = 0.9809 + 0.0025·x	0.4683	0.0314

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
