# Peer review of "Is Biochar from the Torrefaction of Sewage Sludge Hazardous Waste?"

_materials, 2020, doi:10.3390/ma13163544_

Round 1

Reviewer 1 Report

The research conducted in this paper is quite interesting and following the current trends and waste management problems.

Consistency is needed in using the "and" and "&".

The experiments number (no. of variants) in the Abstract is confusing - through further reading of the article, I presume it is 6 variants of temperature and 3 variants of process time, i.e. in total 18 combinations?

Lines 50-52 should be elaborated more - it is not clear what the authors wanted to say. I would suggest inserting "both" in line 52, if that was the intention ("However, only 16 papers include both terms "torrefaction" and "sewage sludge" in title").

The explanation in figures is not needed - it should be inserted in the text.

Since the topic is utilization in agriculture, materials recovery needs to be mentioned. Especially, since the process of torrefaction is determined in this paper as an alternative to energy utilization.

Additionally, small grammar corrections are needed.

However, all in all, the paper is very good and interesting.

Author Response

We would like to thank the Reviewer for comments and suggestions. We addressed all of them. Our responses are included in the attached file.

Reviewer 2 Report

I have gone through the manuscript entitled "Is biochar from the torrefaction of sewage sludge hazardous waste?". This study evaluated the influence of the sewage sludge torrefaction parameters on heavy metal content in biochar, metal leachability, and biochar toxicity, and compared it with raw sewage sludge. Overall, the approach of the study in the manuscript is good and could be useful in the public domain, but the manuscript needs considerable revision to reach the public domain. Authors are suggested to address following comments in order to make the manuscript suitable for publication.

*The abstract should be rewritten by detailing the aim and concept of the manuscript. The abstract should state briefly the purpose of the research, the principal results and major conclusions.

*In the last lines, abstract should be revised with the benefits of the study findings and recommendations as a way forward. 

* Provide significant words which are more relevant to the work in logical sequence as ‘keywords’. Also use keywords which are not present in title.

# Introduction:

* The introduction section is required to be improved. The present introduction is very general and need to be elaborative to explore the actual philosophy to design the experiment. The introduction is insufficient to provide the state of the art in the topic. Hypothesis should be given. How this study is different from the available literature?

The originality and novelty of the paper need to be further clarified. What progress against the most recent state-of-the-art similar studies was made in this study?

*The manuscript does not provide interesting and technically sound discussion; it would be better to use more recent references in discussion.

*Under section, discussion, it is recommended to discuss and explain what should be the appropriate policies based on the findings of this study. Also, the results should be further elaborated to show how they could be used for real applications. 

* Authors are suggested to add discussion by explaining trends in the obtained results along with the possible mechanisms behind the trends.

# Conclusion

Authors are suggested to draw major inferences/primary conclusions first quoting the data/results obtained followed by the secondary conclusions/ recommendations reached through the critical analysis/ investigation of the study

Author Response

(The authors gave the same response as above.)

Reviewer 3 Report

It is a good work for the sustainable management of SS. However, the paper lacks innovation, because many similar works have been published. Some aspects should be taken into account.

    1.The manuscript needs editing by someone with expertise in technical English editing paying particular attention to English grammar, spelling, and sentence structure so that the goals and results of the study are clear to the reader.

  1. What is the phase composition of sewage sludge ?
  2. During the torrefaction process, if there is any chemical reaction, what is the change of the chemical composition and morphology of SS.
  3. The manuscript lacks theoretical analysis in detail.

Author Response

(The authors gave the same response as above.)

Reviewer 4 Report

Major comments:

The study comprised of “biochar from the torrefaction of sewage sludge hazardous waste’’. The authors evaluated the influence of sewage sludge torrefaction parameters (temperature and process time) on heavy metal content in biochar, heavy metals leachability and biochar toxicity and compared it with raw SS. The authors claimed that biochar could be considered as hazardous waste, depending on the feedstock. The results shown in this study are lack of innovation and novelty. One of the major flaws is that the study has been validated with old researches, which makes it difficult to consider the results as unique. The literature review is not strong enough to provide research gaps for this research work. Results and discussions section requires more in-depth discussion and special keenness as it arises a lot of questions regarding the effect of temperature, pressure and other parameters changes. Authors are advised to critically discuss these results and also narrate the limitation of the study. Furthermore, the study fails to include proper practical applications of this experiment in its field. The English language used in the study needs some improvement. Sentences need more clarity and better construction. Therefore, the authors are advised to address the following comments carefully.   

Introduction:

The introduction needs to be more emphasized on the research work with a detailed explanation of the whole process considering past, present and future scope. How the present study gives more accurate results than previous studies? It needs to be strengthened in terms of recent research in this area with possible research gaps. It is strongly recommended to add a recent literature survey about different types of biomass technologies, fuels (fossil and renewable), the emission from conventional energy industry fuels and how these fuels affect current CO2 level and alarming global warming. Research gaps should be highlighted more clearly and future applications of this study should be added. Many questions need to be addressed please see the specific comments.

Specific comments:

  1. Abstract: The authors are advised to write some information about applications of this technology and possible future work briefly. It lacks with strong evidence to support their novelty and innovation.
  2. Page 1, line 33: “leachability varied significantly (p<0.05) from as low. . .” if the results are significant then how come the p-value greater than 0.05?
  3. Page 3, Line 121: The aerobic, anaerobic and anoxic methods used for wastewater treatment, why other processes like biological water treatment, chemical water treatment and sludge water treatment are not useful for this treatment of wastewater? Justify this statement with more recent studies on wastewater treatment: Journal of Environmental Chemical Engineering, 2020; 8(4)104023. Journal of Molecular Liquids, 2020; 313:113494.
  4. Page 4, Line 130: Before experiment SS was dried for 24 h at 105 0 C, what would happen if authors increased the temperature while drying? Please justify with reason.
  5. Page 4, Line 132: ‘’The target temperature is maintained at three retention time variants’’ How the authors maintained the temperature for this variants? What about error analysis? Please explain and add uncertainty analysis of the data obtained.
  6. Page 4, Line 135: The authors are advised to address the reason to ensure inert condition by CO2 during torrefaction process?
  7. Page 4, Line 136: Why heating of reactor began 5 mint after the gas was introduced into the reactor?
  8. Page 4, Line 139-140: This experiment is divided into three retention time variants and six temperature variants under different operational conditions, although it is complicated for the maintenance perspective and other controlling parameters issue. Why authors have not divided it into five retention time variants and three temperature variants to make it simpler?
  9. Authors are advised to make a strong background and discussion with more recent work about different types of fuels and technologies: International Journal of Chemical Reactor Engineering, 2019; 17(11).   Journal of Natural Gas Science and Engineering, 2020; 103313. Waste and Biomass Valorization, 2020 Jul;11(7):3677-709
  10. Page 5, Line 199-200: How temperature and torrefaction duration tends to increase the hydrophobicity? Please elaborate with the mechanism.
  11. Page 6, Line 224-226: Cu content in raw SS is of 33.5mg cu Kg-1m, but in Biochar this Cu content alters from 29.09-41.48mgcu Kg-1d.m for all temperature range, why? Also torrefaction to 60 mint did not significantly affect the Cu content? Elaborate.
  12. Page 11, Line 309-311: Why Cr content increment with time duration of 20, 40 and 60 mint is less than Mn and Zn? Justify with reason.
  13. Page 13, Line 389: If the leachability is in the range….’’According to author the leachability in between 1 and 5mg. Kg-1m considered as hazardous, while on the other hand leachability for calculated Cd range from 1.06 to 1.22mg. Kg-1 d.m, this would also have nature hazardous? Then how this study is novel from other studies?
  14. Page 16, Line 410: The toxicity analysis of Daphnia magna Straus is performed on undiluted water, why used undiluted water, is it possible to use deionized water for this analysis? Elaborate and support with reference.      
  15. Page 16, Line 418: Why toxicity decreases with increasing temperature? Also, explain why different temperatures were used in these retention time variants?
  16. Experimental section: In the Experimental section, the authors need to specify methods, models to use and to discuss ethical issues related to this work, what they intend to do and what was the need of this study clearly. Also needs some focus on describing the effect of process conditional changes.
  17. Conclusion: The authors are advised to write the conclusion in a comprehensive way, it should contain key values, suitability of the applied method, contributions and possible future work.
  18. Figure 1 needs to be revised as the text on the axis is too small and not clear.
  19. The authors are advised to revise references, including the latest references. Please see some suggestions in the specific comments and for the ‘introduction’ section.

Author Response

(The authors gave the same response as above.)

Round 2

Reviewer 2 Report

The authors have addressed all the comments, therefore the manuscript may be published in the present form.

Reviewer 4 Report

The authors have addressed most of the comments; they have also tried to make changes according to the reviewers’ suggestions. After revisions, the quality of the manuscript has been adequately enhanced. Therefore, the manuscript could be considered for the publication in the Journal. However, there are still some editing/ syntax errors present in the manuscript which need to be corrected, hence the publishing team is advised to read the manuscript carefully before publishing.